# Factor Normalization for Deep Neural Network Models

## Abstract

Deep neural network (DNN) models often involve features of high dimensions. In most cases, the high-dimensional features can be decomposed into two parts. The first part is a low-dimensional factor. The second part is the residual feature, with much-reduced variability and inter-feature correlation. This leads to a number of interesting theoretical findings for deep neural network training. Accordingly, we are inspired to develop a new *factor normalization* method for better performance. The proposed method leads to a new deep learning model with two important features. First, it allows factor related feature extraction. Second, it allows adaptive learning rates for factors and residuals, respectively. This leads to fast convergence speed on both training and validation datsets. A number of empirical experiments are presented to demonstrate its superior performance. The code is available at https://github.com/HazardNeo4869/FactorNormalization

## 1 Introduction

In recent decades, the progress of deep learning, together with advances in GPU devices, has led to a growing popularity of deep neural network (DNN) models in both academia and industry. DNN models have been widely used in various fields, such as image classification (Simonyan & Zisserman, 2014; He et al., 2016a), speech recognition (Hinton et al., 2012; Maas et al., 2017), and machine translation (Wu et al., 2016; Vaswani et al., 2017). However, due to their deep structure, most DNN models are extremely difficult to train. The practical training of a DNN model often highly depends on empirical experience and is extremely time consuming. Therefore, a series of effective optimization methods have been developed for fast DNN training.

According to a recent survey paper by Sun et al. (2019), most of the optimization methods with explicit derivatives can be roughly categorized into two groups: the first-order optimization methods and the high-order optimization methods. The widely used stochastic gradient descent (SGD) algorithm and its variants (Robbins & Monro, 1951; Jain et al., 2018) are typical examples of the first-order optimization methods. The SGD algorithm only computes the first-order derivatives (i.e., the gradient) using a randomly sampled batch. By doing so, the SGD algorithm can handle large-sized datasets with limited computational resources. Unfortunately, the practical feasibility of SGD comes at the cost of sublinear convergence speed (Johnson & Zhang, 2013a). For better convergence speed, various accelerated SGD algorithms have been developed. For instance, the popularly used momentum method (Polyak, 1964; Qian, 1999) and the Nesterov Accelerated Gradient Descent (NAG) (Nesterov, 1983; Sutskever et al., 2013) method. Both of them took the information from the previous update gradient direction into consideration. Further improvements include AdaGrad (Duchi et al., 2011), AdaDelta (Zeiler, 2012), RMSprop (Tieleman & Hinton, 2012), Adam (Kingma & Ba, 2014) and others. For a more stable gradient estimation, the stochastic average gradient (SAG) (Roux et al., 2012) and stochastic variance reduction gradient (SVRG) (Johnson & Zhang, 2013b) methods are also developed.

Except for the first-order optimization methods, high-order optimization methods also exist. Popular representatives are the Newton's method and its variants (Shanno, 1970; Hu et al., 2019; Pajarinen et al., 2019). Compared to the first-order methods, high-order methods might lead to faster convergence speed since they take the information of Hessian matrix into consideration.

For example, the Newton's method can have a quadratic convergence speed under appropriate conditions (Avriel, 2003). However, calculating and storing the Hessian matrix and its inverse is extremely expensive in terms of both time and storage. This leads to the development of some approximation methods, such as Quasi-Newton method (Avriel, 2003) and stochastic Quasi-Newton method (Luo et al., 2014). The idea of Quasi or stochastic Quasi Newton method is to approximate the inverse Hessian matrix by a positive definite matrix. For example, DFP (Fletcher & Powell, 1963; Davidon, 1991), BFGS (Broyden, 1970; Fletcher & R, 1970; Donald & Goldfarb, 1970) and L-BFGS (Nocedal & Jorge, 1980; Liu & Nocedal, 1989) methods are popular representatives. Moreover, as a useful technique for fast convergence, various pre-conditioning techniques are also popularly used. (Huckle, 1999; Benzi, 2002; Tang et al., 2009). The basis idea of pre-conditioning is to transform a difficult or ill-conditioned linear system (e.g., $A\theta = b$) into an easier system with better condition (Higham & Mary, 2019). As a consequence, the information contained in the feature covariance can be effectively used (Wang et al., 2019). Other interesting methods trying to extract useful information from the feature covariance also exist; see for example Denton et al. (2015),Ghiasi & Fowlkes (2016), and Lai et al. (2017).

However, to our best knowledge, there seems no existing models or methods that are particularly designed for high-dimensional features with a factor structure. In the meanwhile, ample amounts of empirical experience suggest that most high-dimensional features demonstrate a strong factor type of covariance structure. In other words, a significant amount of the feature variability can be explained by a latent factor with very low dimensions. As a consequence, we can decompose the original features into two parts. The first is a low dimensional factor part, which accounts for a significant portion of the total volatility. The second is the residual part with factor effects removed. This residual part has the same dimension as the original feature. Consequently, it has a much reduced variability. Moreover, the inter-feature correlation is also reduced substantially. To this end, the original learning problem concerning for the high dimensional features can be decomposed into two sub learning problems. The first one is a learning problem concerning for the latent factor. This is relatively simple since the dimension of the factor is very low. The second problem is related to the residual feature. Unfortunately, this is still a challenging problem due to the high dimensions. However, compared with the original one, it is much easier because the inter feature dependence has been substantially reduced. For a practical implementation, we propose here a novel method called factor normalization. It starts with a benchmark model (e.g., VGG or ResNet) and then slightly modifies the benchmark model into a new model structure. Compared with the benchmark model, the new model takes the latent factor and residuals as different inputs. The benchmark model is remained to process the residuals. The latent factor is then put back to the model in the last layer. This is to compensate for the information loss due to factor extraction. By doing so, the new model allows the factor-related features and residual-related features to be processed separately. Furthermore, different (i.e., adaptive) learning rates can be allowed for factor and residuals, respectively. This leads to adaptive learning and thus fast convergence speed.

The rest of this article is organized as follows. Section 2 develops our theoretical motivation with statistical insights. Section 3 provides the details of the proposed new model. Section 4 demonstrates the outstanding performance of the propose model via extensive empirical experiments. Section 5 concludes the article with a brief discussion for future research.

## 2 THEORETICAL MOTIVATION

To motivate our new model, we provide here a number of interesting theoretical motivations from different perspectives. Since the SGD algorithm is a stochastic version of the GD algorithm, we thus focus on a standard GD algorithm in this section for discussion simplicity.

### 2.1 THE GD ALGORITHM

Let $(X_i, Y_i)$ be the observation collected from the $i$-th instance with $1 \leq i \leq N$, where $Y_i$ is often the class label and $X_i = (X_{i1}, ..., X_{ip})^\top \in \mathbb{R}^p$ is the associated $p$-dimensional feature. The loss function evaluated at $i$ is defined as $\ell(Y_i, X_i^\top \theta)$, where $\theta \in \mathbb{R}^p$ is the unknown parameter. Then, the global loss is given by $\mathcal{L}_N(\theta) = N^{-1} \sum_{i=1}^{N} \ell(Y_i, X_i^\top \theta)$. The global gradient is given by

$\dot{\mathcal{L}}_N(\theta) = N^{-1}\sum_{i=1}^N \dot{\ell}(Y_i, X_i^\top\theta)X_i$ and $\dot{\ell}(y,z) = \partial\ell(y,z)/\partial z$. Let $\hat{\theta}^{(t)}$ be the estimator obtained in the $t$-th iteration. Then, the GD algorithm updates the parameter as $\hat{\theta}^{(t+1)} = \hat{\theta}^{(t)} - \alpha\dot{\mathcal{L}}_N(\hat{\theta}^{(t)})$. Here, $\alpha$ is a scalar and is referred as the learning rate (Robbins & Monro, 1951). Assume that $\mathcal{L}_N(\hat{\theta}^{(t)})$ reaches its minimum at $\hat{\theta}$ such that $\dot{\mathcal{L}}_N(\hat{\theta}) = 0$. We then apply Taylor expansion for $\dot{\mathcal{L}}_N(\theta)$ at $\hat{\theta}$. This leads to

$$
\begin{aligned}
\hat{\theta}^{(t+1)} - \hat{\theta} &= \left\{I_p - \alpha\ddot{\mathcal{L}}_N(\hat{\theta})\right\}(\hat{\theta}^{(t)} - \hat{\theta}) + o(\|\hat{\theta}^{(t)} - \hat{\theta}\|^2) \\
&= \mathbb{K}(\hat{\theta}^{(t)} - \hat{\theta}) + o(\|\hat{\theta}^{(t)} - \hat{\theta}\|^2),
\end{aligned}
$$

where $\mathbb{K} = I_p - \alpha\ddot{\mathcal{L}}_N(\hat{\theta})$. We refer to $\mathbb{K}$ as a *contraction operator*, and it plays a very important role in optimization. Intuitively, all the eigenvalues of $\mathbb{K}$ should lie in $(-1, 1)$. Otherwise, the algorithm might not converge numerically.

**Proposition 1** *Assume $\ddot{\mathcal{L}}_N(\hat{\theta})$ to be a positive definite matrix. Let $\lambda_1 \geq \lambda_2 \geq ... \geq \lambda_p > 0$ be the eigenvalues of $\ddot{\mathcal{L}}_N(\hat{\theta})$. To have the GD algorithm converge, we should have $0 < \alpha < 1/\lambda_1$.*

## 2.2 CONDITION NUMBER

By Proposition 1 we know that the learning rate cannot be too large. Otherwise the GD algorithm might not numerically converge. The size of the learning rate is controlled by the largest eigenvalue of the Hessian matrix $\ddot{\mathcal{L}}_N(\hat{\theta})$. The larger $\lambda_1$ is, the smaller the learning rate must be, and the slower convergence speed should be. This problem is particularly serious if the condition number (i.e., $\lambda_1/\lambda_p$) of the Hessian matrix is very large. In that case, the large $\lambda_1$ value forces the learning rate $\alpha$ to be very small. In the meanwhile, other small eigenvalues ($\lambda_j$ for $j \neq 1$) make the convergence speed along the corresponding eigen-directions very slow. Consequently, practitioners should wish the condition number of the Hessian matrix to be as small as possible. On the other hand, as we mentioned previously, most high-dimensional features have a strong factor structure. In other words, the size of the top eigenvalues of the covariance matrix $\Sigma$ is typically much larger than the rest. Consequently, the condition number of its covariance matrix is typically very large. It is then of great interest to investigate: how would this factor structure affect the condition number of the Hessian matrix?

To address this important problem, we evaluate the expected Hessian matrix as $\mathbb{H} = E(\ddot{\mathcal{L}}(\theta)) = E\{\ddot{\ell}(Y_i, X_i^\top\theta)X_iX_i^\top\}$, where $\ddot{\ell}(y,z) = \partial\dot{\ell}(y,z)/\partial z$ stands for the second order derivative of $\ell(y,z)$ with respect to $z$. For illustration purpose, assume $X_i$ is normally distributed with mean 0. Recall that the covariance matrix is $\Sigma$. Define $\tilde{X}_i = \Sigma^{-1/2}X_i$ and $\tilde{\theta} = \Sigma^{1/2}\theta$. We then rewrite $\mathbb{H}$ as $\mathbb{H} = \Sigma^{1/2}E\{\ddot{\ell}(Y_i, \tilde{X}_i^\top\tilde{\theta})\tilde{X}_i\tilde{X}_i^\top\}\Sigma^{1/2} = \Sigma^{1/2}\widetilde{\mathbb{H}}\Sigma^{1/2}$, where $\widetilde{\mathbb{H}} = E\{\ddot{\ell}(Y_i, \tilde{X}_i^\top\tilde{\theta})\tilde{X}_i\tilde{X}_i^\top\}$. Let $A$ be an arbitrary positive definite matrix. Define $\lambda_{\max}(A)$ and $\lambda_{\min}(A)$ be the maximum and minimum eigenvalues of $A$, respectively. We then have $\lambda_{\max}(\mathbb{H}) \geq \lambda_{\max}(\Sigma)\lambda_{\min}(\widetilde{\mathbb{H}})$ and $\lambda_{\min}(\mathbb{H}) \leq \lambda_{\min}(\Sigma)\lambda_{\max}(\widetilde{\mathbb{H}})$. This further suggests that the condition number of $\mathbb{H}$ (i.e., $\mathrm{con}(\mathbb{H}) = \lambda_{\max}(\mathbb{H})/\lambda_{\min}(\mathbb{H})$) satisfies the following inequality,

$$
\mathrm{con}(\mathbb{H}) \geq \mathrm{con}(\Sigma)/\mathrm{con}(\widetilde{\mathbb{H}}), \tag{1}
$$

where $\mathrm{con}(A)$ stands for the condition number of an arbitrary positive definite matrix. Note that $\widetilde{\mathbb{H}}$ is the expected Hessian matrix under an extremely ideal situation, where the input feature follows a standard multivariate normal distribution. This is arguable the most ideal situation for numerical optimization. We thus can reasonably expect that $\mathrm{con}(\widetilde{\mathbb{H}})$ should not be very large in this case. Thus, by (1) we know that $\mathrm{con}(\Sigma)$ should play an important role in determining the $\mathrm{con}(\mathbb{H})$. In other words, the large condition number of $\Sigma$ will affect $\mathrm{con}(\mathbb{H})$. This makes the numerical optimization by a standard GD algorithm extremely difficult, and thus calls for a novel solution.

## 2.3 THE FACTOR LINEAR SUBSPACE

To further motivate the proposed model, we provide some theoretical justification from a different perspective. Assume a standard factor model as $X_i = BZ_i + \mathcal{E}_i$, where $Z_i \in \mathbb{R}^d$ is

a vector with low dimensionality (i.e., $d \ll p$) and $B \in \mathbb{R}^{p \times d}$ is the corresponding factor loading matrix with $d \ll p$. Consider the general loss function $\mathcal{L}(\theta)$. Recall that the global gradient is $\dot{\mathcal{L}}_N(\theta) = N^{-1} \sum_{i=1}^N \dot{\ell}(Y_i, X_i^\top \theta) X_i$. Then, we have $\dot{\mathcal{L}}_N(\theta) = Q_1 + Q_2$, where $Q_1 = B N^{-1} \sum_{i=1}^N \dot{\ell}(Y_i, X_i^\top \theta) Z_i \in \mathcal{S}(B)$, $Q_2 = N^{-1} \sum_{i=1}^N \dot{\ell}(Y_i, X_i^\top \theta) \varepsilon_i$, and $\mathcal{S}(B)$ represents the linear subspace spanned by the column vectors of $B$. Then, the covariance structure of the global gradient can be written as $\text{cov}\{\dot{\mathcal{L}}_N(\theta)\} = \text{cov}(Q_1) + \text{cov}(Q_2)$, where $\text{cov}(Q_1) = B\Sigma_z B^\top / N$, $\Sigma_z = \text{cov}\{\dot{\mathcal{L}}(Y_i, X_i^\top \theta) Z_i\} \in \mathbb{R}^{d \times d}$, $\text{cov}(Q_2) = \Sigma_\varepsilon / N$, and $\Sigma_\varepsilon = \text{cov}\{\dot{\mathcal{L}}(Y_i, X_i^\top \theta) \varepsilon_i\} \in \mathbb{R}^{d \times d}$. It is then of interest to study the relative sizes of $\text{cov}(Q_1)$ and $\text{cov}(Q_2)$ under appropriate metrics.

**Proposition 2** *Assume there exists constant $0 < \tau_{min} < \tau_{max} < \infty$, such that $\tau_{min} < \lambda_{min}(\Sigma_z) \leq \lambda_{max}(\Sigma_z) < \tau_{max}$ and $\tau_{min} < \lambda_{min}(\Sigma_\varepsilon) \leq \lambda_{max}(\Sigma_\varepsilon) < \tau_{max}$. Furthermore, assume that $\lambda_{min}(B^\top B / p) > \tau_{min}$. We then have $\text{tr}\{\text{cov}(Q_1)\} / \text{tr}\{\text{cov}(Q_2)\} \geq \tau_{min}^2 / \tau_{max}$.*

For most DNN models, the parameter dimension $p$ is usually very high. This makes the model structure sufficiently flexible. Accordingly, one might expect the associated GD (or SGD) algorithm should search the entire high dimensional parameter space (e.g., $\mathbb{R}^p$) in a very flexible way. However, the above proposition indicates that the estimated gradient direction is not as flexible as we might expect. In contrast, it always has a significant portion (i.e., $Q_1$) trapped in a very low-dimensional linear subspace $\mathcal{S}(B)$. This brings a positive effect. Since the linear subspace $\mathcal{S}(B)$ is of a very low dimension (i.e., $d$), the overfitting effect due to $Q_1$ becomes negligible. However, this positive effect also comes at a convergence cost. By trapping a significant portion of the gradient direction in $\mathcal{S}(B)$, the capability of a GD algorithm to explore directions other than $\mathcal{S}(B)$ is considerably sacrificed. This problem can be solved if the directions due to the factor ( i.e., $Q_1$) and directions due to the residual features ( i.e., $Q_2$) can be treated separately. This is another important theoretical insight that drives us to develop our new model.

## 3 THE PROPOSED NEW METHOD

Inspired by the theoretical findings discussed in the previous subsection, we propose here a *factor normalization* method. The new method is particularly designed for deep models with high-dimensional features and factor structures. It starts with an important benchmark model (e.g., VGG or ResNet) and can be implemented by the following three important steps. They are factor decomposition, model reconstruction, and adaptive learning, respectively. The details are discussed in the following subsections.

### 3.1 FACTOR DECOMPOSITION

In the first step, we conduct a standard singular value decomposition for the high-dimensional features. By doing so, a low-dimensional factor can be estimated. Accordingly, we can decompose the original features into two parts. The first part is due to the factor and is referred as the factor part. The second part is the original features but has factor effects removed. This part is referred as the residual part (or residual feature). Specifically, the input feature matrix is written as $X = (X_1 ..., X_N)^\top \in \mathbb{R}^{N \times p}$ with $n = \min(N, p)$. Next, we centralize $X$ by its column so that the column mean is 0. We then conduct a singular value decomposition as $X = U\Lambda V^\top = \sum_{i=1}^n \lambda_i u_i v_i^\top$, where $U = (u_1 ..., u_n) \in \mathbb{R}^{N \times n}$ and $V = (v_1 ..., v_n) \in \mathbb{R}^{p \times n}$ with $U^\top U = V^\top V = I_n$ and $I_n \in \mathbb{R}^{n \times n}$ stands for an identity matrix. Moreover, $\Lambda = \text{diag}(\lambda_1 ..., \lambda_n)$ is a diagonal matrix with $\lambda_1 \geq \lambda_2 \geq ... \geq \lambda_n \geq 0$. Define $V_d = (v_1 ..., v_d)$. We then estimate the latent factor by $\hat{Z} = X V_d$, where $\hat{Z} \in \mathbb{R}^{N \times d}$ represents the estimated factor part. Last, we use a linear regression to calculate the residual feature matrix as follows $\hat{\mathcal{E}} = X - \hat{Z}\hat{B}$: where $\hat{B} = (\hat{Z}^\top \hat{Z})^{-1} \hat{Z}^\top X \in \mathbb{R}^{d \times p}$ is the estimated factor loading matrix. By doing so, the original feature matrix $X$ is decomposed into two parts: the factor part $\hat{Z}$ and the residual part $\hat{\mathcal{E}}$.

### 3.2 MODEL RECONSTRUCTION

In the second step, we modify a given benchmark model (e.g., a ResNet) into a new one for better performance. More specifically, consider a standard DNN model with (1) an input layer with

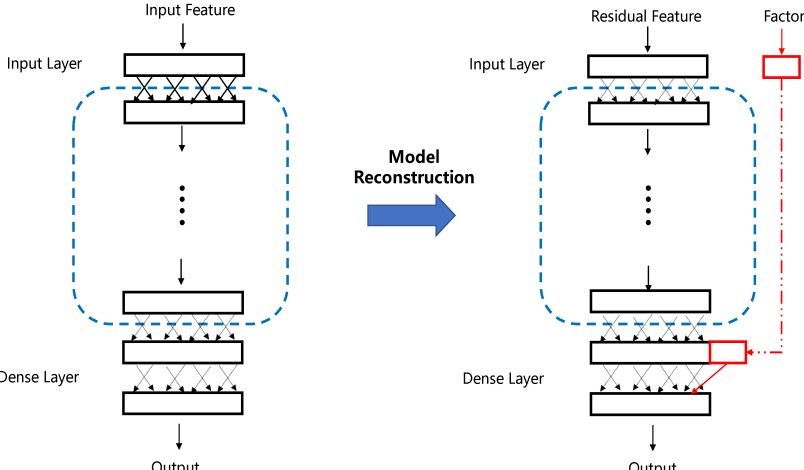

Figure 1: A graphic illustration of MODEL RECONSTRUCTION for factor normalization. The rectangles represent the feature maps in a DNN. The blue dashed rectangle represents the latent layers between the input and the last dense layer. The left panel shows an original DNN structure with the original input feature. The right panel shows a modified DNN structure with factor and residual features. As one can see, the factor and residual feature are treated separately. The latent factor is reinserted back into the model before the last dense layer for output.

original feature $X$; (2) an output layer, which is often of a dense type structure, and (3) sophisticated latent layers between the input and output layers. We then replace the original feature of the input layer (i.e., $X$) with the residual feature (i.e., $\hat{\mathcal{E}}$). Next, let the residual feature be fully processed by the sophisticated latent layers until the output layer. Before the DNN model constructs the last dense layer for output, we put the estimated latent factor (i.e., $\hat{Z}$) back to the output layer to compensate the information loss due to factor extraction.

This leads to a new model structure with two interesting characteristics. The high-dimensional residual term is still processed by the sophisticated benchmark model. However, the low-dimensional latent factor is not much processed. This interesting structure is particularly desirable for the following reasons. First, for most cases, we find one dimensional factor seems already good enough. In this case, the latent factor is univariate. Then, its any nonlinear transformation (e.g., by a deep neural network) should make little difference from the original factor, in terms of the information contained. Thus, there is little need to further conduct sophisticated nonlinear transformation for the univariate latent factor. In contrast, the residual form remains to be high dimensional. Thus, sophisticated nonlinear transformation is still very useful for excellent feature extraction. Second, we want to use different learning rates for factor and residuals for adaptive learning and thus better performance. If the factor is put in some intermediated layer, the practical implementation of adaptive learning by TensorFlow programming could be extremely and unnecessarily complicated. This explains why the latent factor is set as the input of the last layer. Figure 1 gives a more intuitive explanation.

## 3.3 ADAPTIVE LEARNING

As we mentioned before, with this modified model, the convergence speed is still very slow if we adopt a standard SGD algorithm. For a reliable convergence, a small learning rate should be used for the parameters associated with the factor because factor variability is large. In contrast, much larger learning rates should be used for the parameters associated with the residual features, otherwise the convergence speed could be very slow due to its small variability. This suggests

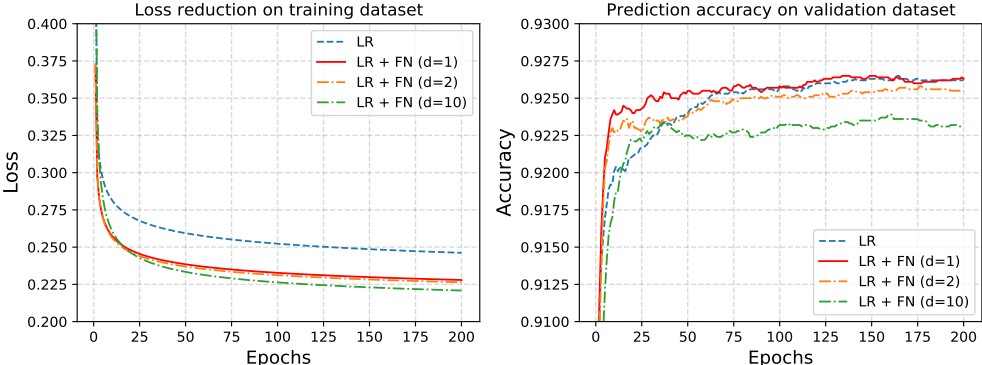

Figure 2: LR stands for logistic regression and FN stands for factor normalization. The left panel shows the training loss on MNIST images of the logistic regression model and FN models optimized by SGD. The right panel shows the prediction accuracy on the validation dataset. Different numbers of factors are tested (i.e., $d = 1, 2, 10$).

that adaptive (i.e., different) learning rates should be used for the factor and residual. Specifically, assume the learning rate used in a standard SGD is $\alpha$. We then set the learning rate for the factor associated learning rate to be $\alpha_j = \alpha \lambda_1^2 / \lambda_j^2$ for $j = 1..., d$. Recall that $\lambda_j$ is the $j$th largest singular value of $X$ (column centralized). In contrast, the learning rates used for the residual associated learning rates are set to be the same as $\alpha_\varepsilon = \alpha \lambda_1^2 / \lambda_{d+1}^2$. The consequence is that the learning rates used for the latent factor become much smaller than those of the residual features. It should be noted that the adaptive learning rates used here are different from those used in AdaGrad or Adam Duchi et al. (2011); Kingma & Ba (2014). The AdaGrad and Adam methods adjust their learning rates dynamically for each iteration and adaptively for each dimension in the training process.

## 4 EXPERIMENTS

To empirically demonstrate the proposed FN model, we conducted various experiments on different models, including logistic regression, multilayer fully connected neural networks and deep convolutional neural networks. All the results are based on optimal settings of initial learning rate using grid search and $1/\sqrt{t}$ decay rate. The experiments were run on a Tesla K80 GPU with 11GB memory.

### 4.1 LOGISTIC REGRESSION

We start with a simple logistic regression model and evaluate the proposed model on the MNIST dataset. This is a classification task with 10 classes, 60,000 instances for training, and 10,000 instances for validation. The input feature is a $28 \times 28$ pixel matrix. To implement the logistic regression, we reshape the input feature into a 784 dimension vector. Subsequently, the FN models with various factor dimensions ($d = 1, 2, 10$) are used to reconstruct the baseline model. The SGD optimizer is used to optimize both the baseline model (i.e., LR) and FN models. The detailed results are given in Figure 2. For comparison purposes, all the models are trained for a large number of epochs so that its performance on a validation data are fully converged. Specifically, a batch size of 200 with 200 epochs are adopted. The left panel of Figure 2 reports the training loss for various models. We find that the FN models consistently outperform their benchmark counterparts, since the loss curves of the FN models are always below the baseline model. The right panel of Figure 2 displays the results of accuracy. We can see the accuracy curves of FN models reach the plateau earlier than the baseline models. Additionally, it seems that there is little difference in the performance of FN models with different factor dimension $d$. We are then inspired to fix $d = 1$ for the subsequent experiments.

## 4.2 MULTILAYER NEURAL NETWORKS

We next consider a more complicated multilayer neural network model. The proposed model has two fully connected hidden layers with 1,000 neurons. The Rectified Linear Unit (ReLU) transformation is used in each hidden layer for activation (Kingma & Ba, 2014). The settings of the experiment are almost the same as those in subsection 4.1 except for the following differences. First, we only consider the FN model with $d = 1$. Second, for a comprehensive comparison purpose, we consider four different optimizers, they are SGD, NAG, Adagrad and Adam, respectively. The detailed results are given in Figure 3. The top panel of Figure 3 reports the training loss in log scale for the four

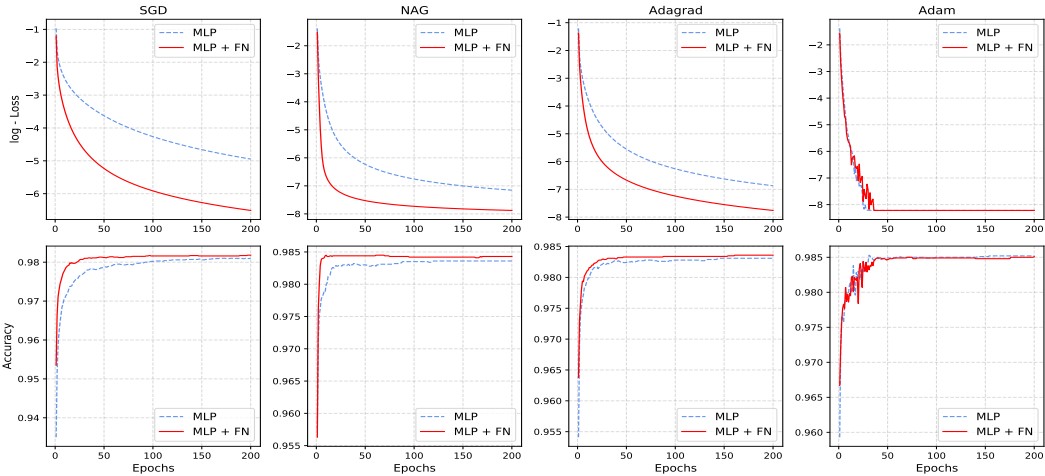

Figure 3: The top panel is the training loss in log scale for baseline model (blue dash line) and FN model (red solid line) with four different optimizers. The bottom panel is the validation accuracy for baseline model (blue dash line) and FN model (red solid line) with different optimizers.

optimization methods on the baseline model and FN model. We find that for each optimizer, the FN model always achieves smaller training loss than the baseline model. This difference is more obvious in the cases of SGD, NAG and Adagrad. The bottom panel of Figure 3 reports the prediction accuracy on validation dataset. We find that for each optimizer, the accuracy of FN model is higher than that of the baseline model.

## 4.3 CONVOLUTIONAL NEURAL NETWORKS

Convolutional neural networks (CNNs) are the most popular models used in image classification and object detection tasks. The main difference between CNNs and multilayer neural networks is that CNNs contain many convolutional and pooling layers, which provide efficient feature extraction and lead to excellent prediction performance. However, CNNs usually have complicated model structures with a large number of parameters. Here, we consider two classical CNN models. They are AlexNet (Krizhevsky et al., 2012) and ResNet50 (He et al., 2016b). The dataset used here are CIFAR10 (Krizhevsky & Hinton, 2009) and CatDog released by Kaggle in 2013. For illustration purpose, the results of AlexNet on CIFAR10 are summarized in Figure 4, while the results of ResNet50 on CatDog are given in Figure 5. Similar to the experiment in subsection 4.2, a total of four baseline optimization methods are also evaluated. They are SGD (Robbins & Monro, 1951), NAG (Bengio et al., 2013), AdaGrad (Duchi et al., 2011) and Adam (Kingma & Ba, 2014).

The left panel of Figure 4 reports the time consumption (includes the time of conducting SVD) required by different optimization methods (i.e., SGD, NAG, AdaGrad and Adam) to reach the optimal prediction accuracy for baseline model and FN model. The optimal prediction accuracy is set as the accuracy that is 1% lower than the optimal accuracy of baseline models. We find that for all the four optimization algorithms (i.e., SGD, NAG, AdaGrad and Adam), the time is substantially reduced for FN models. The reduction is significant in the cases of SGD and Adam. The right panel of Figure 4 compares training loss in log scale. For a fair comparison, 250 epochs are given

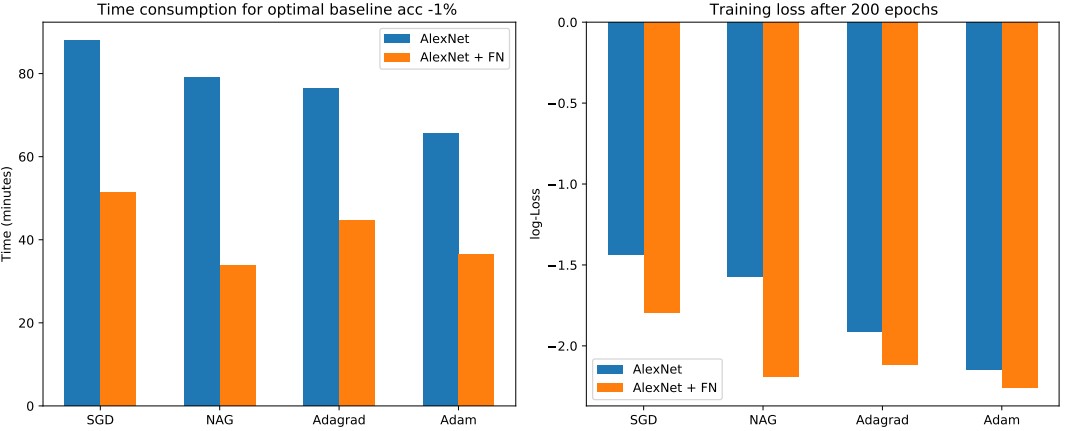

Figure 4: Results of AlexNet on the CIFAR10 dataset. A benchmark accuracy is set as the accuracy that is 1% lower than the optimal accuracy of baseline model. The left panel shows the time cost for each optimization algorithm to obtain the benchmark accuracy between baseline model and FN model. The right panel shows the training loss in log scale after 200 epochs for the four optimization algorithms between baseline model and FN model.

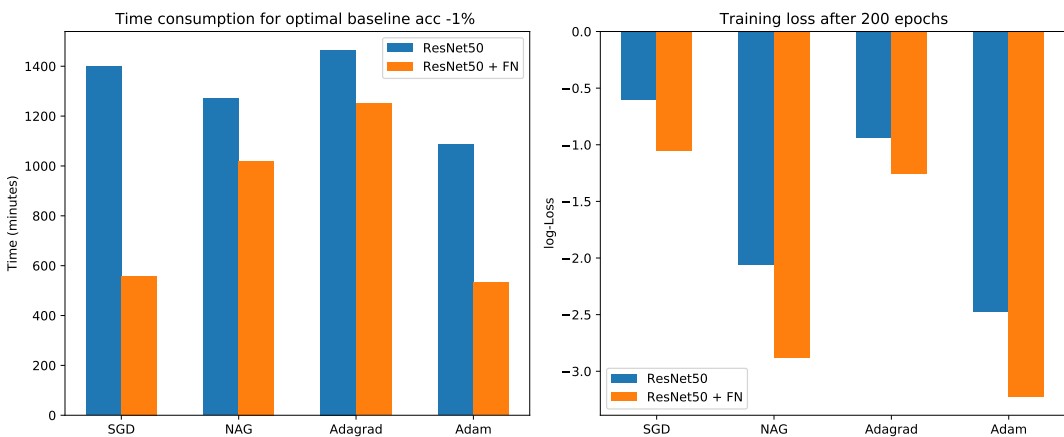

Figure 5: Results of ResNet50 on the CatDog dataset. A benchmark accuracy is set as the accuracy that is 1% lower than the optimal accuracy of baseline model. The left panel shows the time cost for each optimization algorithm to obtain the benchmark accuracy between baseline model and FN model. The right panel shows the training loss in log scale after 200 epochs for the four optimization algorithms between baseline model and FN model.

for different optimizers. We find that the training loss can be relative lower for the baseline model at the end of long time training. Overall, we find that the FN model can significantly improve the performance of a baseline model in terms of time cost, training loss, and prediction accuracy. The results of ResNet50 given in Figure 5 are qualitatively similar.

## 5 CONCLUSION

In this paper, we propose a novel *factor normalization* method for fast deep neural network training. The idea was inspired by the fact that many DNN models involve high dimensional features and these features are often of a strong factor structure. The proposed method has three key components. First, it decomposes a high dimensional input feature into two parts. One is the factor part with low

dimensionality, and the other is the residual part. Second, it modifies a given DNN model slightly so that the effect of the latent factor and residual feature can be processed separately. Last, to train a modified DNN model, a new SGD algorithm is developed. It allows adaptive learning rates for the factor part and the residual part.

To conclude this article, we present here a number of interesting topics for future study. First, our current FN model conducts factor decomposition on the input feature. In fact, even if different components of the input feature are completely independent of each other, the resulting feature map after a number of convolutional layers might still have a strong factor structure. In this case, whether factor decomposition should be conducted for the feature maps is a problem worth studying. Second, the current FN model does not insert the estimated factors back into the DNN models until the last layer. What would happen if the estimated factors were also inserted into earlier layers? This is another problem of great interest.

ACKNOWLEDGMENTS

This research is supported by the National Natural Science Foundation of China (No.71702185, No.71873137, No.11971504, No.71532001, No.11525101, No.71332006), Beijing Municipal Social Science Foundation (No.19GLC052), National Statistical Science Research Project (No.2020LZ38), Ministry of Education Focus on Humanities and Social Science Research Base (Major Research Plan 17JJD910001) and China's National Key Research Special Program (No. 2016YFC0207704), Consulting Research Project of Chinese Academy of Engineering(2020-XY-30).

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
