# OpenReview forum: "Factor Normalization for Deep Neural Network Models"
_ICLR.cc/2021/Conference — Reject_

### Official Review · AnonReviewer3 · 2020-10-24
**Interesting simple idea, but requires more scholarship.**

**Rating:** 5
**Confidence:** 4

**Review:**

Review of "Factor normalization for DNNs".

The paper makes an observation that datasets used for training many deep neural nets exhibit a strong factor structure, i.e. have a small number of dominant principal components explaining most of the variance.  If we were to remove the dominant factors, the residuals would have much weaker correlation structure and allow faster DNN convergence with SGD.   The paper proposes to separate the dominant factors and the residuals,  train the original DNN on residuals with a much faster SGD learning rate,  and then recombine with a shallow small NN learned on the dominant factors trained using its own (slower) learning rate.

First the positive aspects:  this is an interesting idea, and I haven't seen it commonly used in practice for DNNs.  While mathematical analysis is conducted for linear models with GD and is fairly straightforward, it nicely illustrates the main issues, and the hope is that the linear intuition continues to apply to deep NNs.  Experimental results suggest that indeed convergence rate can be improved on several datasets.   In terms of criticisms,  there is very limited scholarship of related ideas that have been used both for linear models and for DNNs, in particular (a) various factor-based models that already exist,  (b) preconditioning of linear systems,  (c) neural nets trained on some other sort of residuals -- e.g. laplacian pyramid DNNs .  Another criticism -- is that there is no discussion of how to do large-scale PCA / factor analysis for high-dimensional image data arising in say modern DNN image classification pipelines, and its computational cost,  as simple numpy.linalg.svd won't work.  The paper also claims that strong factor structure (with a small number of dominant components) is prevalent in modern ultra-large scale DNN applications -- I would like to see some references / supporting evidence beyond just computer vision.

Overall in my opinion the paper needs to consider the context of related works, and has a few other correctable issues, but I would certainly encourage the authors to continue to improve it.

Additional details:

1)  Prior work -- that should be cited / contrasted with your approach:
(a) Since a substantial part of the paper analyzes linear models,  it's important to mention that factor structure has been long exploited in various ML / stats works.  For example factor models, and principal component regression (PCR) attempt to focus the modeling power on the principal components. In other applications (e.g. in financial modeling) one can assume that factors components are less predictable and focus the modeling effort on the residuals. Recent work by Alex Smola et al,  "Deep factors for forecasting" has looked at the deep-NN instantiation of this idea.  Basic autoencoders or bottlenecks used in DNN architectures also attempt to capture the low-dimensional structure.  These are all different from what you're doing, but hinge on the same basic concept -- so providing a discussion of your work in context of related work would be important.
b)  There is a long history of using preconditioners for gradient-based and other iterative solvers of linear systems.  In particular there are some low-rank preconditioners for accelerating convergence of linear systems:  Nicholas Higham, et. al, "A new preconditioner that exploits low-rank approximations to factorization error".  There is also work on applying preconditioners specifically for SGD, e.g. see works by Michael Mahoney.
c)  For some domains a low-pass filter, or some other low-resolution model can serve to replace PCA or factor models. For example Laplacian pyramids have been widely used in image processing to separate the dominant modes from details.  There are existing DNN approaches based on laplacian pyramids:
e.g. "Deep laplacian pyramid networks for fast an accurate super-resolution".

2)  You mention that "none of these optimization methods has considered the covariance structure".. related to the Hessian.  There is a substantial effort to develop second-order methods for stochastic gradient descent, in particular at ICML 2020 there was a workshop on this topic.  "Beyond first order methods in ML".   http://www.wikicfp.com/cfp/servlet/event.showcfp?eventid=102387&copyownerid=156008

3) In section 2.1. -- it's worth mentioning explicitly that you're specifically analyzing linear regression under gradient descent.  as you look at Loss(y, theta' X).   It's also worth explicitly mentioning that the condition number of the Hessian plays a crucial role in convergence rate of GD.   You talk about the top eigenvalue, where the bottom one in your example is fixed at 1,  but it's worth mentioning the condition number.

4) Sec. 3.1. How do you conduct 'standard principal component analysis' for ultrahigh dimensional features.  This is a computationally tricky problem, "standard" methods won't work.   How do you decide on the dimension -- i..e. number of factors to keep?

5) Is the "time consumption"  including the time to do PCA?

6) Can you give some references claiming strong factor structure in several DNN applications in ultra-high dimensions?  I do not contest that this is the case -- but it would be useful to have supporting evidence.  What number/fraction of factors is typically required in these applications to capture a nontrivial fraction of variance?

7) Sec. 2.1. Using lower-case kappa for a matrix is strange,  I initially assumed it's a scalar.  Maybe use another capital letter.

8) While the paper is mostly pretty readable, there are various small issues with english language (from stylistic to grammar) use e.g. "In fact ample amounts of empirical evidence" --> "ample empirical evidence", e.t.c. There are typos in references, e.g.  Zeiler,   "Computer ence, 2012". What is that?

---

### Official Review · AnonReviewer1 · 2020-10-28
**Interesting phenomena but not easy to understand**

**Rating:** 4
**Confidence:** 3

**Review:**

Summary:
In this paper, a learning method that accelerates the training of DNN is proposed. Given an input X, the proposed method decomposes X as X = BZ + E where BZ is a low-rank approximation of X and E is the residual term. E is used as an input of DNN and Z is used as an additional feature of the input of the last layer. Experiments using MNIST and CIFAR10 show that the proposed method accelerates the speed to reduce the training loss.


Detailed comments:
The observed phenomena --- the proposed method accelerates the learning speed of SGD --- is quite interesting. I haven't noticed existing studies reporting such findings. However, I feel "why" parts are not clearly explained in this paper. For example, I have the following questions.
- The analysis in Section 2 is based on a shallow network model. How can we apply this analysis to a deep network model?
- What kind of intuition is behind the architecture design (Fig 1)? Why the residual term should place as the input of the first layer and the main term as the input of the last layer, rather than e.g. some intermediate layer?

Also, I feel the experiments are not convincing enough. The main motivation of this paper is that "the ultrahigh dimensional features can be decomposed into two parts". However, the dimensions of MNIST and CIFAR10 are not quite ultrahigh (28^2=784 and 32^2=1024).  Experiments with more high-dimensional data such as ImageNet would be necessary to convince the research concept.

---

### Official Review · AnonReviewer2 · 2020-10-28
**experiments not very convincing**

**Rating:** 4
**Confidence:** 3

**Review:**

The paper describes a training scheme based on decomposing input features into two parts which have different training dynamics: a low rank "factor feature" computed using PCA on the raw features, and a high rank "residual".  The former is processed  by a very shallow network, while the latter passes through the full network, and parameters for each are updated with different learning rates.  Experiments show that the proposed algorithm can speed up wall-time to  a certain accuracy on MNIST and CIFAR10 classification, across several neural network architectures and optimizers.

Pros:
- Simple and straightforward proposal, easy to understand, and seems to lead to improved accuracy/reduced loss early in training.
- Experiments cover several datasets and architectures spanning very simple to very deep, which is nice to see.

Concerns:

- Very "mathy" presentation in Sec. 2 is very dense and difficult to follow, and only seems to motivate the method in the special cases of a very shallow linear regression model, where the gradient is easily related to eigenvalues of the input features $X$ (and assuming a "strong factor structure", which has been shown to be reasonable for natural images.)  It's not clear why these results would necessarily generalize to very deep and highly nonlinear networks such as AlexNet.  Perhaps this is the motivation for only processing "factor features" with an extremely shallow (albeit still nonlinear) network? If so, this should be more clearly explained in the text.

- The experiments are not very convincing.
 - The baselines trained with vanilla SGD or other optimized (Adam, etc.) are not clearly explained.  Do they use the same feature factorization as SGD+FN or do they consist of only the raw features passed into the full network (i.e. the left side of Figure 1)?  If the latter, the comparisons are not totally fair as the input features and networks differ.
 - Models are not trained until convergence, but only for a fixed amount of time, or until a fixed (but not very high) accuracy is reached.  How does the proposed scheme affect final accuracy/loss at convergence?  (E.g., I'd expect simple logistic regression on MNIST to achieve over 95% accuracy)   Is it strictly a training speedup, while converging to the same performance as baselines?  Or is it only suitable when operating within a limit training time budget?
  - This issue is illustrated in the training curves in Fig 3 (MLP on MNIST), where the validation accuracy for the proposed SGD+FN appears to be plateauing faster than the baseline SGD curves.
 - Furthermore, the speedup over NAG or Adam on CIFAR10 using  ResNet50 isn't very large.
 - All training comparisons are measured only as a function of wall-time.  But the training hardware is not explained.
 - Missing obvious baselines of training separate models on either the factor features or residual alone.  At least for MNIST I'd expect decent performance to be obtainable from the factor features alone.  Maybe the residual is mostly irrelevant to the task?

Overall I feel that the paper is not ready for publication at this time since the experimental validation is incomplete and does not fully explore the benefits and potential downsides of the proposed method.

Other comments:

- Sec 3.3: "Adaptive learning".  If I'm understanding correctly, the learning rates are defined based on the eigenvalues of $X^T X$, but are then kept fixed.  So, unlike e.g. AdaGrad or Adam,  the algorithm is not really adaptive in the sense that the learning rate is kept constant throughout training.

- Sec 3.3: "a standard SGD algorithm cannot be used to train a DNN model".   This seems like an unnecessarily strong statement given how commonly SGD is used for DNN training.

- The linked code does not appear to include resnet experiments

---

### Official Review · AnonReviewer4 · 2020-10-29
**The paper proposed to project input to factor (low-dimensional) and residual features (high-dimensional) to improve neural network training.**

**Rating:** 4
**Confidence:** 3

**Review:**

**Strong points**

The paper provides a very detailed theoretical analysis of motivation.

**Weak points**

Analysis in section 2 is based on linear regression, but the proposed method is based on deep models.

The proposed method is more similar to feature extraction instead of a training method.

Many models in the chart are not fully converged. It is important to compare convergence speed but also the final accuracy. I think they are not converged, because the training and testing curve is perfectly smoothing without any fluctuations.

Comparison in chart 3 needs improvement. Since the two models have different input signals and model structure, they inherently need different hyper-parameters (not only learning rate) for best performance. Only comparing them with the same learning rate may not adequate to prove the significance of the proposed method.

---

### Decision · Program_Chairs · 2021-01-07
**Final Decision**

**Decision:**

Reject

**Comment:**

The authors proposed to pre-process the original input features into a low dimensional term and its corresponding residual term via SVD. The paper empirically demonstrated the neural networks trained on such factorized exhibit faster convergence in training. Several issues of clarity were addressed during the rebuttal period by the authors.

However, the reviewers still felt that there were some remaining fundamental issues with the paper,

1)  The motivation is not echoed in the experiments, namely most of the experiments on CIFAR and CatDog dataset using a low dimensional factorization of d=1 which is trivial and often part of the whitening preprocessing.

2) The proposed factorization via SVD will be difficult to scale up to high dimensional features, large training sets and higher d >> 1.

3) The empirical experiments show a marginal improvement in the training speed, especially in the image recognition tasks, yet there seems an early plateau in test performance when compared to the baselines.

4) The theoretical analysis in Section 2 studied linear models. Yet, the rest of the paper focuses on non-linear neural networks. It is difficult to see the connection between the analysis and the rest of the paper.

Thus, I recommend rejection of the paper at this time as the current version of the paper needs further development, and non-trivial modifications, to be broadly applicable.